# Effect of Indigenous and Introduced Arbuscular Mycorrhizal Fungi on Growth and Phytochemical Content of Vegetatively Propagated *Prunus africana* (Hook. f.) Kalkman Provenances

**DOI:** 10.3390/plants9010037

**Published:** 2019-12-25

**Authors:** Yves H. Tchiechoua, Johnson Kinyua, Victoria Wambui Ngumi, David Warambo Odee

**Affiliations:** 1Department of Molecular Biology and Biotechnology, Pan African University Institute for Basic Sciences, Technology and Innovation (PAUSTI), P.O. Box 62000, Nairobi 00200, Kenya; 2Department of Biochemistry, Jomo Kenyatta University of Agriculture and Technology (JKUAT), P.O. Box 62000, Nairobi 00200, Kenya; johnsonkinyua@jkuat.ac.ke; 3Department of Botany, Jomo Kenyatta University of Agriculture and Technology (JKUAT), P.O. Box 62000, Nairobi 00200, Kenya; vngumi@jkuat.ac.ke; 4Biotechnology Laboratory, Kenya Forestry Research Institute (KEFRI), P.O. Box 20412, Nairobi 00200, Kenya; dwodee@gmail.com; 5Centre for Ecology & Hydrology, Bush Estate, Penicuik EH26 0QB, UK

**Keywords:** arbuscular mycorrhizal fungi, growth parameters, indigenous AMF, phytochemical compounds, *Prunus africana*

## Abstract

*Prunus africana* bark contains phytochemical compounds used in the treatment of benign prostatic hyperplasia and prostate cancer. It has been shown that this plant establishes association with arbuscular mycorrhizal fungi (AMF). AMF are involved in nutrient uptake, which may also affect plant growth and secondary metabolites composition. However, there is no information regarding the role of AMF in the growth and phytochemical content of *P. africana*. A pot experiment was carried out to assess the response of 8 months old vegetatively propagated *P. africana* seedlings inoculated with indigenous AMF collected from Mount Cameroon (MC) and Mount Manengumba (MM) in Cameroon, Malava near Kakamega (MK) and Chuka Tharaka-Nithi (CT) in Kenya. Mycorrhizal (frequency, abundance and intensity), growth (height, shoot weight, total weight, number of leaf, leaf surface) and phytochemical (total phenol, tannin and flavonoids) parameters were measured three months after growth of seedlings from two provenances (Muguga and Chuka) with the following inoculation treatments: MK, CT, MC, MM, non-sterilized soil (NS) and sterilized sand as non-inoculated control. Results showed that seedling heights were significantly increased by inoculation and associated with high root colonization (>80%) compared to non-inoculated seedlings. We also found that AMF promoted leaf formation, whereas inoculation did not have any effect on the seedling total weight. AMF inoculum from MM had a higher tannin content, while no significant difference was observed on the total phenol and flavonoid contents due to AMF inoculation. Pearson’s correlation was positive between mycorrhizal parameters and the growth parameters, and negative with phytochemical parameters. This study is the first report on the effect of AMF on the growth and phytochemical in *P. africana*. Further investigations are necessary to determine the effect of single AMF strains to provide better understanding of the role of AMF on the growth performance and physiology of this important medicinal plant species.

## 1. Introduction

*Prunus africana* (Hook. f.) Kalkman is an endemic African species distributed in almost 20 countries from western, central, eastern to southern Africa [1,2]. It is an important multipurpose medicinal tree whose bark, stem, roots and leaves have been used traditionally for many decades in Africa [3,4]. Modern use of the bark includes the treatment of benign prostatic hyperplasia (BPH) [5,6] and prostate cancer [7]. The pharmaceutical and medicinal properties of *P. africana* have led to its overexploitation throughout Africa, leading to concerns on the long-term sustainability of harvesting and the conservation of this species. As a result of overexploitation, *P. africana* was listed in 1995 as endangered Species under appendix II of CITES [8]. Due to its economical, pharmaceutical and medicinal importance, *P. africana* has gained a lot of interest by NGOs, governments, scientists and farmers to improve its domestication. Consequently, a number of studies have been reported on the regeneration through seed propagation [9] and vegetative propagation [9,10,11] of *P. africana*. However, previous studies have not addressed the role of soil microorganisms, especially arbuscular mycorrhizal fungi (AMF) on vegetatively propagated *P. africana*. Yet the species is known to associate with AMF in natural habitats [12,13]. Arbuscular mycorrhizal fungi are ubiquitous soil microorganisms that form a symbiotic association with ~71% of terrestrial plants [14]. AMF belong to the sub-phylum Glomeromycotina and the phylum Mucoromycota [15]. AMF specificity and host preference have been shown to be variable by a number of studies. For instance, Campos et al. [16] showed that wheat could preferentially establish association with a specific community of AMF, whereas Torrecillas et al. [17] observed variable trends in AMF specificity with herbaceous plant species in the semiarid Mediterranean prairies specificity or preference in AMF-plant association is driven by the efficiency of the balance carbon and nutrient supplying. However, a large number of AMF are ubiquitous with low or no specify to plant species [18]. These soil microorganisms help host plants to absorb mineral nutrients such like phosphorus (P), nitrogen (N), sulphur (S), potassium (K), calcium (Ca), iron (Fe), copper (Cu), and zinc (Zn); in return they benefit from the host plants by acquiring organic carbon in form of glucose and lipid [19,20]. In addition to direct nutrient benefits, AMF have been reported to offer ecosystem services, including resistance to diseases like root infections caused by pathogenic fungi [21,22], increased tolerance to drought [23] and improving the soil structure with the aggregate formation [24,25,26]. These benefits contribute to the growth and adaptation of plants in diverse environmental conditions. Arbuscular mycorrhizal fungi can improve secondary metabolite contents in medicinal plants by improving plant nutrients status and/or altering the hormonal balance of the plants [27]. Several studies have been carried out to investigate the effects of AMF inoculation on the concentration of secondary metabolites compounds. Flavonoids concentration was shown to have increased in the bark of *Libidibia ferrea* after being inoculated with *Claroideoglomus etunicatum* and *Gigaspora albida*; similarly, the concentration of total tannins was shown to be higher when inoculated with *Acaulospora longula* compared to the non-inoculated control plants [28]. Arbuscular mycorrhizal fungi species can also have different effects in the host; for instance, Copetta et al. [29] showed that *Glomus mossea* increased the concentration of alpha-terpineol—an essential oil from *Ocimum basilicum*—compared to *Gigaspora margarita*, *Gigaspora rosea* and non-inoculated control plants. The aim of this study is to assess the effect and provenance of AMF on the growth and the phytochemical content of vegetatively propagated *P. africana* seedlings under glasshouse conditions.

## 2. Materials and Methods

### 2.1. Soil Samples, Trap Cultures and AMF Inoculum Production

The soil samples were collected from the rhizosphere of *P. africana* in Mount Cameroon (MC) (04°08′34.3″ N and 09°07′21.0″ E, 2280 m asl) and Mount Manengumba (MM) (05°01′50.8″ N and 09°49′31.7″ E, 1968 m asl) in Cameroon, and in Chuka forest in Tharaka-Nithi (CT) (0°17′45.57″ N and 37°36′52.85″ E, 1620 m asl) and Malava forest in Kakamega (MK) (0°27′57.57″ N and 34°52′8.55″ E, 1615 m asl) in Kenya. The diameter at breast height (dbh) of selected trees was above 30 cm. At each of the four cardinal points, at about 20 cm distance from the bark, approximatively 50 g of rhizosphere soil were collected at a depth of 20 cm from the soil surface. Samples from each tree were pooled. Sampled soils (100 g) from each source were placed on top of sterilized sand (500 g; autoclaved at 121 °C for 1 h) and then covered with another layer of sterilized sand (200 g) in a 2 L container. Trap cultures were established in pots by sowing 4 to 5 surface sterilized seeds of *Sorghum bicolor* (sorghum) and *Vigna unguiculata* (cowpea) into the top layer of sterilized sand. The trapping was monitored for 3 months with regular watering, followed by another month of stopping watering. After four months, AMF spores were isolated using the wet sieving and sucrose density gradient centrifugation methods [30], then the different morphotypes were identified based on the identification key in INVAM platform (https://invam.wvu.edu/) [31] (Figure 1). Representative healthy isolated spores of AMF were used to produce AMF inoculum using the single-spore inoculation technique [32]. *Sorghum bicolor* seeds were surface sterilized using 70% ethanol for 2 min, following by 1% sodium hypochlorite solution for 3 min. Then, the seeds were transferred to Petri dishes containing water agar 0.8% and kept in the oven at 30 °C for 3 days. The germinated seeds were then placed in a folded filter paper and a single healthy spore of AMF isolated and sterilized was placed on the root. AMF inoculum produced was pooled based on the sample site and therefore, four inocula were produced hereafter referred to as Malava-Kakamega (MK), Chuka-Tharaka-Nithi (CT), Mount Cameroon (MC) and Mount Manengumba (MM). Each AMF inoculum was a mixture of AMF’s spores, root fragments of trap plants and sterilized sand substrate. Inoculum MK, CT, MC and MM were then considered as Indigenous AMF inoculum.

### 2.2. Production of Vegetatively Propagated P. Africana

Healthy and fresh stem cuttings of *P. africana* were collected from juvenile trees to produce leafy stem cuttings, in two different environments conditions (Table 1), the Muguga forest and the Chuka forest. A non-mist propagator described by Leakey et al. [33] was used for the production of *P. africana* seedling. The leaf free end of stem cuttings of *P. africana* was soaked into auxin Indole-3-Butyric Acid (IBA) 200 mg/L concentration, for 1 min [9]. The treated cuttings were planted in the non-mist propagator for the production of *P. africana* seedlings free of mycorrhizal colonization with an average day and night temperature, respectively at 31 °C and 15 °C. After 3 months, the stem cuttings developed roots and leaves. The rooted seedlings were introduced into experimental plastic pots containing sterilized (autoclaved at 121 °C for 1 h) mixture of soil (collected from the producer of *P. africana* seedlings at Kenya Forest Research Institute (KEFRI)) and sand, then placed in the greenhouse at KEFRI. The transferred seedlings were maintained in the greenhouse for 3 additional months for acclimation.

### 2.3. AMF Inoculation Procedure

The experiment was conducted using a two-factor completely randomized block design containing six inoculation treatments and two treatments sources of vegetatively propagated *P. africana* seedlings (Muguga forest and Chuka forest). The inoculation treatments were as follows: MK, CT, MC and MM as described above (section Soil samples and trap cultures production), non-sterilized soil (NS) Muguga nursery soil (Table 2, see also [36]) used for the propagation of *P. africana* seedlings, and sterilized sand (autoclaved at 121 °C for 1 h) as non-inoculated control treatment and all treatments were done in triplicate. For mycorrhizal inoculation, 500 g of inoculum produced, containing infected root fragments, hyphae and spores were introduced in pots containing vegetatively propagated *P. africana* seedlings. The experiment was carried out in the greenhouse for a period of three months from October 2018 to January 2019, with a photoperiod of 12 h, and average day and night temperatures of 25 °C and 14 °C, respectively.

### 2.4. Measurement of Mycorrhizal Colonization Parameters, Growth and Phytochemical Contents Parameters

After 3 months, seedlings were harvested by carefully uprooting from the substrate. Approximately 2 g of fresh roots were subsampled from each seedling and stored in 50% ethanol to evaluate the mycorrhizal colonization. After washing the fine roots with distilled water, they were cleared using 10% KOH at room temperature for 24 h, treated with 30% H_2_O_2_ for 30 min, washed, acidified with 10% HCl at room temperature for 15 min and stained with 0.05% Trypan blue in lactoglycerol at 121 °C for 5 min [37]. Thirty root fragments of approximately 1 cm were analyzed under the microscope. The following mycorrhizal parameters were analyzed: frequency (F), AMF’s intensity in roots system (AIRS), AMF’s intensity in the root fragment (AIRF), abundance of arbuscules in the root system (AARS), and abundance of arbuscules in the root fragment (AARF). These parameters were evaluated as described by Trouvelot et al. [38] using the MYCOCALC software (www.dijon.inra.fr/mychintec/Mycocalc-prg/download.html). To quantify seedling growth, the number of leaves was counted, the average leaf surface area using the fast and accurate method [39]. Seedling height, and dry shoot and the total weight were also determined after drying at 50 °C to constant weight in a drying oven. The phytochemicals, namely tannin content, total flavonoids and phenols were measured. Condensed tannins were assayed using vanillin-hydrochloric acid method as described by Price et al. [34,40]. Extraction was done using 4% HCl in methanol using a shaker (Labortechnik KS 250b, Germany). After separation using a refrigerated centrifuge (Kokusan, Type H-2000C, Japan) at 4500 rpm for 10 min at 25 °C, extraction was repeated on the residue from the first extraction using 1% HCl in methanol and standards were prepared using catechin hydrate. Absorbance for all prepared solutions was read at 500 nm and tannin content calculated as percent catechin equivalent (CE) using the standard calibration curve. Total flavonoids were determined using aluminium chloride colorimetric method [41] with 0.3 mL of 5% sodium nitrite solution added to a mixture of 1 mL of plant extract and 4 mL of distilled water. Then 10% aluminium chloride was added to a mixture and after 5 min, 2 mL of 1 M sodium hydroxide was added, and the volume made up to 10 mL with distilled water. Absorbance was measured at 415 nm using UV-Vis spectrophotometer (Shimadzu model UV–1601 PC, Kyoto, Japan). The amount of total flavonoids was calculated from Calibration curve of standard prepared from quercetin. Total phenol compounds in samples were determined using the Folin–Ciocalteu method [42]. Fifty mL of methanol was added to 5 g of ground sample and shaken for 3 h, then kept for 72 h and filtered. After being centrifuged for 10 min at 150 rpm at 25 °C, 1 mL of the supernatant was filtered, then mixed with 2 mL of Folin–Ciocalteu and vortex. After 2 h, absorbance of the mixture was determined at 765 nm.

### 2.5. Statistical Analyses

All the statistical analyses were conducted using R studio 3.5.3 [43] software, excepted where indicated. Shapiro test was performed to check the normality of the data. Leaf count data were square root (sqrt) transformed. Data were analyzed by analysis of variance (ANOVA), with the means of treatments being compared using R version 3.5.3. When data was not following the assumption of normality, a Kruskal–Wallis test was performed, followed by Pairwise comparisons using Wilcoxon rank sum test as a post-hoc test at *p* ≤ 0.05. We transformed AMF colonization parameters before applying Pearson’s correlation test for the relationship between AMF colonization parameters, phytochemical parameters and plant growth parameters.

## 3. Results

### 3.1. Mycorrhizal Colonization Parameters

AMF colonized all samples examined, except the non-inoculated control plants as revealed by Trypan blue staining. In the colonized roots, vesicles, internal and external hyphae could be observed (Figure 2). AMF colonization parameters were generally higher on seedlings propagated from Muguga forest than those from Chuka forest. However, significant differences were only observed in arbuscules abundance in a root fragment (AARF), with 10.5% and 29.6% respectively in Chuka forest and Muguga forest (Figure 3). In General, AMF colonization frequency was significantly higher in inoculated treatments with 72%, 70%, 68% and 51% respectively for CT, MK, MC and MM compared to non-sterilized Muguga nursery soil (NS) at 38% and the non-inoculated control treatments having 0%. (Figure 4).

### 3.2. Growth Parameters of P. Africana Seedlings

We observed that seedlings from leafy stem cuttings collected at Muguga forest had higher growth performance, irrespective of treatment, when considering height (cm), leaf surface area (cm^2^), shoot weight (g) and total weight (g), with 4.367 ± 0.26, 5.604 ± 0.558, 0.129 ± 0.025 and 0.375 ± 0.037, respectively (Table 3). ANOVA and Tukeys test showed significant difference at *p* < 0.05 (Figure 5). The different treatments did not have any significant effect on shoot weight and total weight. However, we also observed variable growth response to inoculation. Seedlings inoculated with CT, MK, MC and MM had significantly higher growth respectively than non-inoculated control treatment, while only seedlings inoculated with CT, MK and MM had greater leaf surface area than non-inoculated treatment. There was no significant difference in shoot and total dry weight among the treatments (Figure 6).

### 3.3. Phytochemical Content of P. Africana Seedlings

In contrast to the growth performance, vegetatively propagated seedlings from cuttings collected in Chuka forest showed higher phytochemical contents (tannins, flavonoids and phenols) than the seedlings from Muguga Forest (Table 4; Figure 7). On the other hand, seedlings inoculated with inoculum MM produced significantly higher tannin content, while total phenol and flavonoids content was significantly high with inoculum CT compared to other inoculum treatments. However, there was no significant difference in flavonoids content among CT, MM and the non-inoculated control. Similar observation was made on total phenol content, with no significant effect between CT and MM inoculum (Figure 8).

Pearson correlation analyses revealed a positive correlation between mycorrhizal parameters and all the growth parameters in general, except for the total weight (Figure 9). In contrast to growth parameters, we observed a negative correlation between mycorrhizal parameters and phytochemical content in the *P. africana* seedlings (Figure 9).

## 4. Discussion

This study is the first attempt to investigate the role of AMF on the growth and phytochemical content of *P. africana*. The soil substrate used by the farmers to raise *P. africana* seedlings is usually collected where there are no *P. africana* species. Therefore, a comparative effect of soil used as an inoculum containing AMF propagules from the rhizosphere soil of different populations of *P. africana* will improve the management of this important plant in the nursery. In our study, we found that mycorrhizal and non-mycorrhizal *P. africana* seedlings collected from Muguga forest, appeared to have a higher growth parameter than the seedlings from Chuka forest. This could be explained by the different environmental conditions of the sources of the stem cutting (Table 1). These conditions might have contributed to the natural adaptation of seedlings collected at Muguga forest where the experiment was undertaken and shown better growth parameters, as opposed to seedlings raised from cuttings collected from Chuka forest, located approximately 194 km from the experiment site. On the other hand, seedlings collected from Chuka forest contained a higher concentration of tannins, flavonoids and total phenol content compared to the seedlings collected in Muguga forest. Kadu et al. [1] research revealed a genetic variability among *P. africana* species found in Africa and specifically in Kenya. Consequently, the genetic diversity among *P. africana* population can possibly lead to a different response in the interaction with the environment. However, we also observed that all *P. africana* seedlings roots from both provenances were colonized by AMF (more than 70% in most cases) when they were inoculated with AMF inoculum, except the control treatment. In [12,44], it was concluded that *P. africana* was a mycotrophic species due to the presence of AMF structure in the root system, respectively in Ethiopia and Cameroon. We observed that all the AMF inoculum, had a positive effect on the growth parameters of *P. africana*’s seedlings in nursery, compared to the control treatment where no AMF inoculum was added. However, we observed an increase in shoot weight and total weight, albeit not significant (*p* = 0.64 and *p* = 0.77, respectively). In contrast, seedlings height and leaf surface were significantly higher (*p* = 0.0083 and *p* = 0.0059, respectively) compared to the control, when the seedlings were inoculated with AMF (Figure 5). This could be due to the early stage association between AMF and seedlings, which may involve an unbalanced bidirectional exchange of carbon transfer and nutrients during the early stages of growth [45]. AMF inoculum had a significant and positive effect on leaf surface area and number. Chen et al. [46] have reported that AMF can have from positive to neutral and even negative effect to the host, depending on the functional specialization between the two partners involved in the interaction. The non-sterilized (NS) soil used by *P. africana* seedlings producers’ had a negative effect on leaf surface area and total biomass production notwithstanding the observed mycorrhization. Indigenous AMF inoculum have been shown to have higher benefits compared to the exotic. Cheng et al. [47] study the effect of indigenous versus exotic *Rhizoglomus intraradices* on *Poncirus trifoliata* (trifoliate orange). These authors found that indigenous AMF had a higher effect on plant growth compared to the exotic strains. In our study, AMF inoculum were produced from AMF spores collected in the rhizosphere soil of *P. africana* in the natural habitat, and therefore, could be considered as indigenous, whereas NS treatment containing AMF spores, were exotic to *P. africana*. The bark of *Prunus africana* has been reported to be used in the treatment of benign prostatic hyperplasia (BPH) [5,6,48] and recently, its potential to be used as Chemoprevention and Chemotherapy of Prostate cancer has been explored [7]. The bark of the tree has also been reported as a control agent for the fungi and bacterial infections [49]. The listed properties of *P. africana* barks are attributed to different types of phytochemical compounds. Our study targeted three phytochemical compounds namely tannins, flavonoids and phenols; tannins in *P. africana* have been reported to be having antimicrobial and antioxidant properties [50]. We observed no significant difference between treatments on the tannins content, except the treatment MM which shows a higher concentration. Flavonoids like many other polyphenol compounds are essentially plants and fungi secondary metabolites products. Previous reports and/or studies have mentioned the presence of this group of phytochemical compounds in *P. africana* extracts [5,51]. Also known to have antioxidant activities, phenol, especially flavonoids play a key role in management of prostate conditions [50,52,53,54]. The flavonoid and total phenol contents were significantly higher in CT compared to other mycorrhizal treatments. Other studies, such as Geneva et al. [55] have reported a decrease of total phenol and flavonoids content in *Salvia officinalis*. We obtained similar findings in our study; the Pearson’s correlation showed a negative correlation between mycorrhizal parameters and phytochemical content (Figure 9). However, other research has shown increased total phenol and flavonoid in *Amburana cearensis* [56] and *Viola tricolor* [57] inoculated with AMF. A number of researches have shown a variation of phytochemical content during the different phases of the plant growth, from their increasing [58] to their alteration [59]. Therefore, different values of tannins, flavonoids and total phenols in *P. africana* could be either increased or decreased during the growing.

## 5. Conclusions

This study showed that AMF inoculum produced with indigenous AMF spores from *P. africana* rhizosphere soil, significantly enhanced the growth parameters (height, leaf surface area, number of leaves and the total weight) of *P. africana* seedlings in nursery. However, the response in phytochemical content of *P. africana* seedlings was variable. We also found that *P. africana* seedlings collected in Muguga forest had better growth parameters compared to the seedlings collected in Chuka forest. We used inoculum produced from a mixture of AMF spores. Therefore, further studies will be required to identify the AMF species in the inoculum, then to assess the effect of single AMF species on the phytochemical of *P. africana* seedlings at various stages of growth. This study is the first report on the role of AMF on *P. africana* seedlings, and we believe that these preliminary results will guide the design of new studies especially using single spore with experimentation in the greenhouse as well as in the field with a longer period of observation.

## Figures and Tables

**Figure 1 plants-09-00037-f001:**
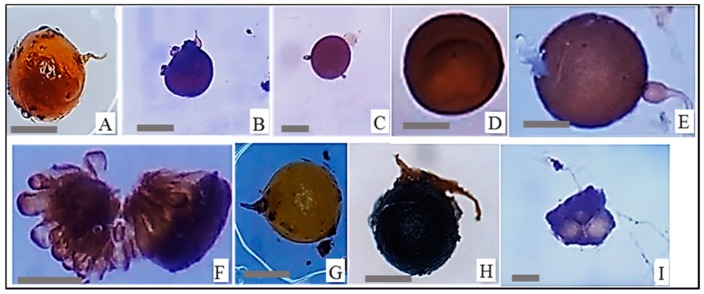
Representative of arbuscular mycorrhizal fungi (AMF) spores isolated and morphologically identified using the identification key of INVAM platform (https://invam.wvu.edu/), from the rhizosphere soil of *P. africana* after 3 months of trap culture. (**A**,**G**) *Scutellospora* sp. (scale bars = 100 μm), (**B**,**C**,**E**) *Gigaspora* sp. (scale bars = 75, 50 and 120 μm respectively), (**D**) *Acaulospora* sp. (scale bar = 100 μm), (**F**) *Sclerocystis* sp. (scale bar = 150 μm), (**H**) *Entrophospora* sp. (scale bar = 100 μm), (**I**) *Glomus* sp. (scale bar = 50 μm).

**Figure 2 plants-09-00037-f002:**
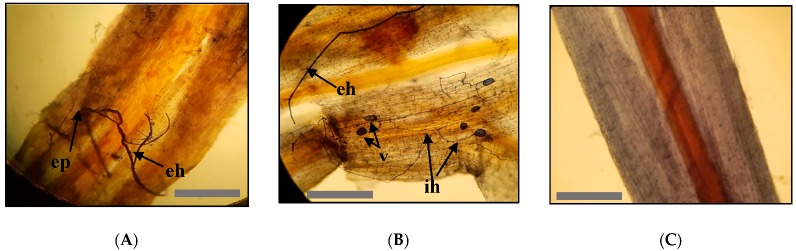
Arbuscular mycorrhizal fungi assessment in roots of *P. africana* stained with Trypan blue. (**A**,**B**) Colonized root fragment from inoculated mycorrhizal treatment showing external hyphae-eh, internal hyphae-ih, entry point-ep and vesicles-v. (**C**) Non-colonized root of non-inoculated (control) treatment seedling. Scale bars = 200 μm.

**Figure 3 plants-09-00037-f003:**
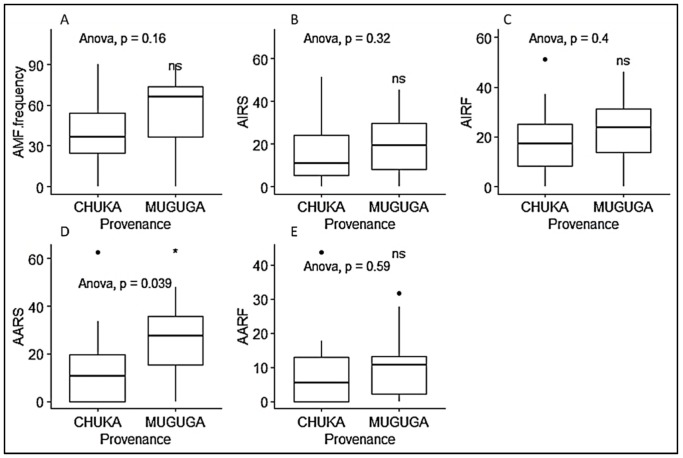
Boxplots representing AMF structure in *P. africana* seedlings roots based on their provenance (Chuka or Muguga). (**A**) AMF frequency; (**B**) AMF intensity in *P. africana* roots system: AIRS; (**C**) AMF intensity in the root fragment of *P. africana* seedlings (AIRF); (**D**) Arbuscules abundance in the roots system of *P. africana* seedling (AARS); (**E**) Arbuscules abundance in the root fragment of *P. africana* seedling (AARF). Data are median with 25% and 75% quartile (box) and non-outlier range (whisker). Statistical significance: ns (*p* > 0.05); * (*p* ≤ 0.05); Black dot: outlier.

**Figure 4 plants-09-00037-f004:**
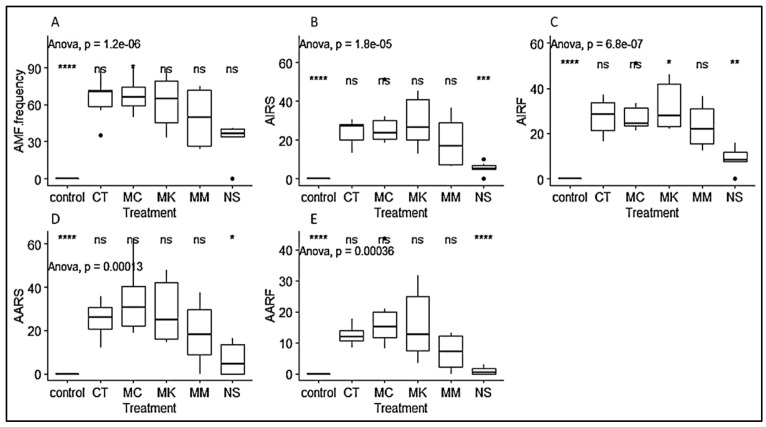
Boxplots representing AMF structure in *P. africana* seedlings roots based on treatments. (**A**) AMF frequency; (**B**) AIRS; (**C**) AIRF; (**D**) AARS; (**E**) AARF. CT: Chuka Tharaka-Nithi, MC: Mount Cameroon, MK: Malava Kakamega, MM: Mount Manengumba, NS: Non-sterilized. Data are median with 25% and 75% quartile (box) and non-outlier range (whisker). Statistical significance: ns (*p* > 0.05); * (*p* ≤ 0.05); ** (*p* ≤ 0.01); *** (*p* ≤ 0.001); **** (*p* ≤ 0.0001); Black dot: outlier.

**Figure 5 plants-09-00037-f005:**
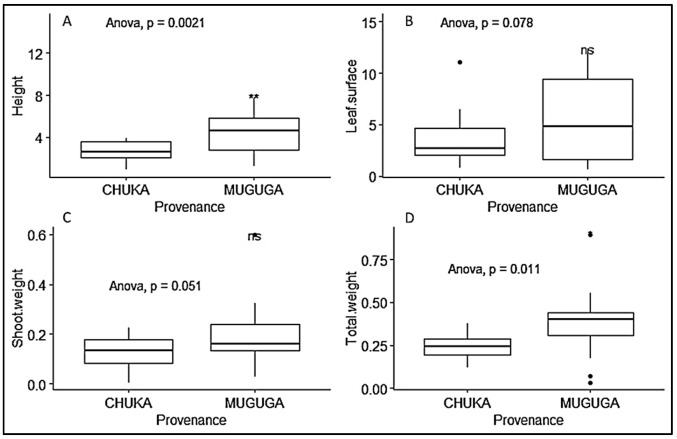
Boxplots representing the effect of provenance on *P. africana* seedlings growth in nursery. (**A**) seedlings height measurement; (**B**) seedlings leaf surface area; (**C**) seedlings shoot weight; (**D**) seedlings total weight. Data are median with 25% and 75% quartile (box) and non-outlier range (whisker). Statistical significance: ns (*p* > 0.05); * (*p* ≤ 0.05); ** (*p* ≤ 0.01); Black dot: outlier.

**Figure 6 plants-09-00037-f006:**
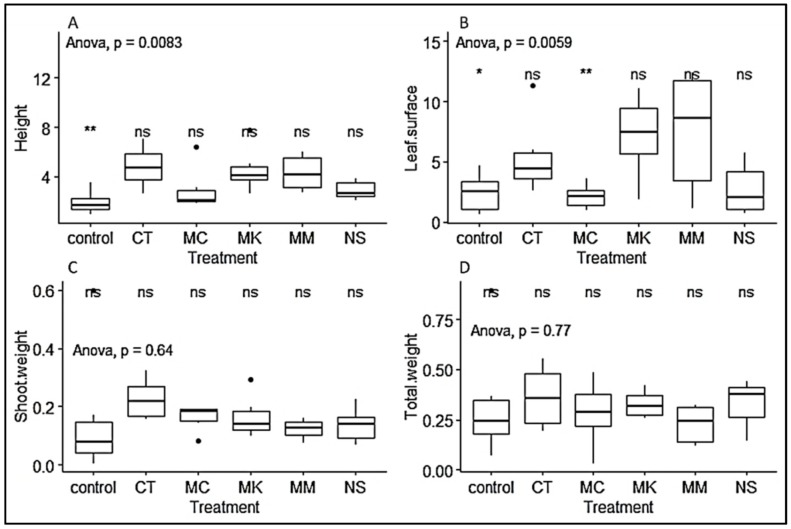
Boxplots representing the effect of different treatments used on *P. africana* seedlings growth in nursery. (**A**) seedlings height measurement; (**B**) seedlings leaf surface area; (**C**) seedlings shoot weight; (**D**) seedlings total weight. Data are median with 25% and 75% quartile (box) and non-outlier range (whisker). Statistical significance: ns (*p* > 0.05); * (*p* ≤ 0.05) ** (*p* ≤ 0.01); Black dot: outlier.

**Figure 7 plants-09-00037-f007:**
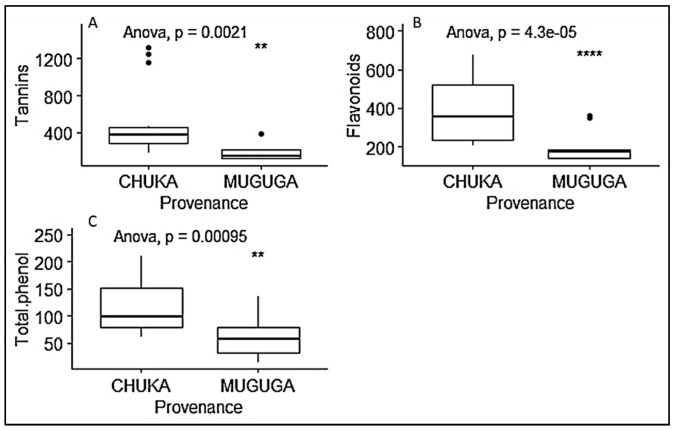
Boxplots representing the effect of provenance on *P. africana* seedlings phytochemical content in nursery. (**A**) tannins content; (**B**) flavonoids content; (**C**) total phenol content. Data are median with 25% and 75% quartile (box) and non-outlier range (whisker). Statistical significance: ** (*p* ≤ 0.01); **** (*p* ≤ 0.0001); Black dot: outlier.

**Figure 8 plants-09-00037-f008:**
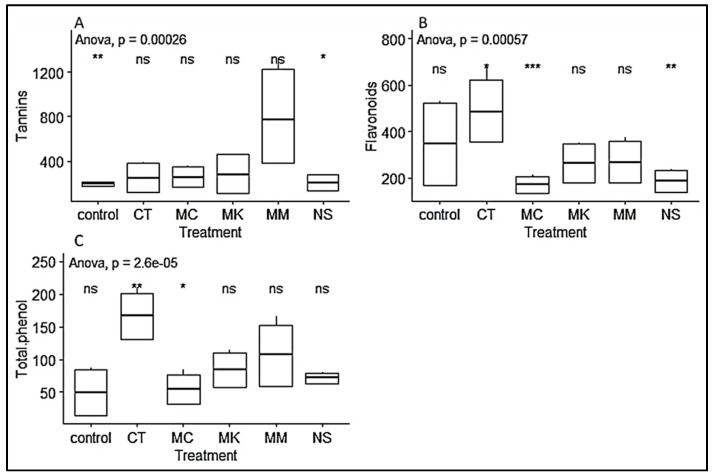
Boxplots representing the effect of different treatments on *P. africana* seedlings phytochemical in nursery. (**A**) tannins content; (**B**) flavonoids content; (**C**) total phenol content. Data are median with 25% and 75% quartile (box) and non-outlier range (whisker). Statistical significance: ns (*p* > 0.05); * (*p* ≤ 0.05); ** (*p* ≤ 0.01); *** (*p* ≤ 0.001); Black dot: outlier.

**Figure 9 plants-09-00037-f009:**
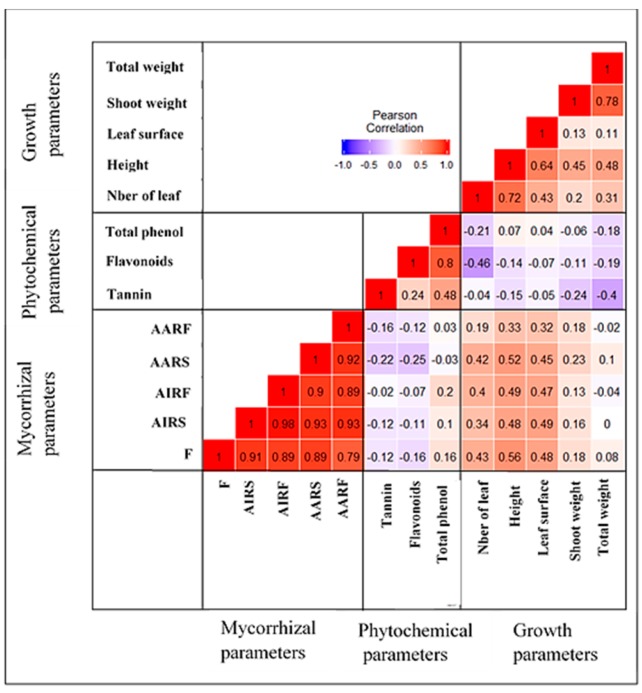
Pearson’s correlation between the different parameters (note: F, frequency; AIRS, Intensity in roots system; AIRF, intensity in the root fragment; AARS, abundance in the root system; AARF, abundance in the root fragment).

**Table 1 plants-09-00037-t001:** Environmental characteristics of the *P. africana* stem cuttings provenance.

Provenance	Characteristic			
	Location	Altitude	Annual Rainfall and Modality	Soil Parameters
Muguga forest [34]	Long: 36°38′ E, 36°37′ E Lat: 01°13′ S, 01°12′ S	2070 masl	990 mm; Bimodal rainfall (March to May and October to December)	Dominated by clay-loam (6.2–6.5)
Chuka forest [35]	Long: 37°19′ E, 37°36′ E Lat: 0°11′ S,0°19′30″ S	2600–4000 masl	1500–2500 mm; Bimodal rainfall (March to June and October to December)	Red clay soil, Nitisols, Cambisols, Andosols

**Table 2 plants-09-00037-t002:** Chemical characteristics of Muguga nursery soil.

pH (H_2_O)	EC mS/cm	C%	N%	P ppm	K ppm	Mg ppm	Ca ppm	Mn ppm	Zn ppm	Cu ppm	Fe ppm
7.1	0.079	1.82	0.26	45	475	395	4633	45	6.7	1.2	189

**Table 3 plants-09-00037-t003:** Mean value of growth parameters of vegetatively propagated *P. africana.*

Provenance	Growth Parameters	Mean	Std. Error
Muguga forest	Log10 Number of leaves	2.572 ^a^	0.075
	Height (cm)	4.367 ^a^	0.260
	Leaf surface area (cm^2^)	5.604 ^a^	0.558
	Shoot weight (g)	0.195 ^a^	0.025
	Total weight (g)	0.375 ^a^	0.037
Chuka forest	Log10 Number of leaves	1.939 ^b^	0.075
	Height (cm)	2.683 ^b^	0.260
	Leaf surface area (cm^2^)	3.502 ^b^	0.558
	Shoot weight (g)	0.129 ^b^	0.025
	Total weight (g)	0.244 ^b^	0.037

Note: ^a, b^ represent the significance differences between plant provenances at *p* ≤ 0.05.

**Table 4 plants-09-00037-t004:** Mean value of growth parameters of vegetatively propagated *P. africana* per provenance.

Provenance	Phytochemical Parameters	Mean	Std. Error
Muguga forest	Tannin (mg.100 g)	188.892 ^b^	5.789
	Flavonoids (mg.100 g)	193.355 ^b^	2.248
	Total phenol (mg.100 g)	62.229 ^b^	0.834
Chuka forest	Tannin (mg.100 g)	482.905 ^a^	5.789
	Flavonoids (mg.100 g)	386.385 ^a^	2.548
	Total phenol (mg.100 g)	116.606 ^a^	0.834

Note: ^a, b^ represent the significance differences between plant provenances at *p* ≤ 0.05.

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
