# Peer review of "Effect of Indigenous and Introduced Arbuscular Mycorrhizal Fungi on Growth and Phytochemical Content of Vegetatively Propagated Prunus africana (Hook. f.) Kalkman Provenances"

_plants, 2019, doi:10.3390/plants9010037_

Round 1

Reviewer 1 Report

The topic of your research is extremely interesting. The manuscript is well written although there are some minor flows that you should revise before publication. Repetition of the experiments more than once is really important for the robustness and scientific soundness of the results. I believe that you should repeat your experiments again and extent the observation time from seedling to the maturation stage of the trees. However, for the content of this paper you have satisfied replicates in each of your treatments and your results are very interesting. Please see the specific comments bellow.

Specific comments

Line 62: “…including resistance to diseases…” such as? Please give an example.

Line 84: can you please tell the exact procedure/protocol for soil sampling? From what depth did you collected samples? What distance from the bark? How old were the trees? Etc

Line 85: So, we need to hypothesize that the indigenous AMF that you used are not species specific since you were able to culture them on Sorghum bicolor. If that is the case you have to add 3-4 lines in the section of introduction and discuss briefly that there are AMF which are general and AMF which are species specific.

Line 117: how many replicates per treatment?

Line 175: any reason for the observed difference?

Lines 186 and 193: The denoted statistical differences through letters is a bit old fashioned and a bit frustrating. There are other more appropriate methods to denote the differences and I would strongly suggest to change this in order to improve your graphs.

Line 205: I believe that you need more time of observation in order to observe differences on shoot weights and total weights.

Line 211: as the comment above.

Line 215: as lines 186 and 193.

Line 240: Figure 8- As previous comment. Please see comment for lines 186-193.

Lines 255-256: OK! Environmental differences could explain the difference at some extent, but can you tell us what the differences between the two forests are. You haven’t mentioned any environmental differences between the two forests.

Lines 294-295: So? How does this correlate with seedling growth?

Lines 300-301: OK, but have you thought that the phytochemical content could be altered during the growth of the tree? Mycorrhizal effects should be observed during the growth of the tree (from seedling to mature) in order to estimate the phytochemical content. Do you think that you should extend your experiments and collect more information on this before you claim that mycorrhiza had no effect on phytochemical content?

Author Response

Please see the attachment for response to Reviewer 1 Comments

Reviewer 2 Report

Minor revision

In this article, the authors studied the effect of indigenous and introduced arbuscular mycorrhizal fungi  on growth and phytochemical content of vegetatively propagated Prunus africana (Hook. f.) Kalkman provenances.

The topic is interesting but there are some comments to the authors:

Please, check the title. It is better to remove abbreviation AMF and extra points (Effect of indigenous and introduced arbuscular mycorrhizal fungi (AMF) on growth and phytochemical content of vegetatively propagated Prunus africana (Hook. f.) Kalkman provenances.) Check, please, in general, the compliance of the submitted material (including the list of references) with the requirements of the journal.

Author Response

Please see the attachment for response to Reviewer 2 Comments 

Reviewer 3 Report

In the presented manuscript „Effect of indigenous and introduced ARBUSCULAR MYCORrhizal Fungi (AMF) on growth and phytochemical content of vegetatively propagated Prunus africana (Hook. f.) kalkman Provenances.” authors tried to evaluate the growth and chemical parameters of inoculated and uniculated seedlings of Prunus africana. The importance of mycorrhiza for growth and development of plants is a hot topic for science, agriculture and ecology. Therefore such studies are generaly on interest.

On the other hand, quiet little can be concluded by the present work;

The mycorrhizal fungi used for inoculation were not characterized, the only known information is a natural source, where the organisms were collected. It seems to be quiet surprising that the inoculation generally led to increase of height and leaf surface, but no significant increase of shoot weight nor seedlings total weight was observed. Is there some explanation for that? This should be addressed in discussion. Although some significant differences were presented between the inoculated and uninoculated plants in their content of tannins, flavonoids or total phenolics, the correlation between the mycorrhiza and phytochemicals seems to be very weak (lines 233-234). Moreover, the used methods of measurements of flavonoids and total phenolics are rather a general estimations, allowing just preliminary conclusions.

The conclusions (lines 298-305) correspond well with the results presented in the manuscript. Up to my opinion, the presented results are a good foundation, but further studies would be necessary to increase the relevance of the manuscript to be suitable for publication in the MDPI Plants.

Author Response

Please see the attachment response to Reviewer 3 Comments

Reviewer 4 Report

As mentioned by the authors, this study describes for the first time the effect of AMF on the growth and phytochemical content in Prunus africana, a medicial endemic African tree species. The authors carried out pot experiments with vegetatively  propagated P. africana seedlings from two locations: Muguga and Chuka and inoculated them respectively with indigenous AMF collected from different locations: MC, MM, MK and CT in Kenya as well nonsterile soil: NS. Mycorrhizal, plant growth and phytochemical content parameters were measured three months after growth from two Provevavces respectively. A positive correlation has been shown between mycorrhizal parameters and all the plant growth parameters in general. This study also showed that, AMF inoculum significantly enhanced the plant growth parameters of P. africana while the response in phytochemical content of P. africana seedlings was variable. Also the P. africana seedlings from Muguga had better growth parameters compared to those from Chuka forest.

In general, I find this present study quite interesting and application orientated as its future implementation might contribute to the rescue of the endangered tree, P. africana from overexploitation throughout Africa. The manuscript is in general well written with some minor clarifications.

However I have some concern on the experiment set up in order to make a more clear observation and conclusion. Considering that in nature many different species and strains coexist in the same field, it is crucial to include AMF mixtures for detecting possible synergistic effects and functional complementarities among them, leading to a further selection of the best AMF combinations. With this idea, while planning the present pot experiments, it would have been better to include one more setup with mix of all the AMFs (MC, MM, MK and CT) used and check its correlation to plant parameters as well as Phytochemical parameters.

Secondly, including a phylogenetic analysis on the molecular diversity of the AMFs used in this study would have helped to understand in deep this particular AMF-plant relationship in a similar way done by Nzweundji et al., 2015, as mentioned in your reference list.

Minor clarifications:

Line 83: sampled soil mixed with sterilized sand...in what ratio?

Figure 1: scale bar missing

Line 172/173 “as revealed Trypan blue staining” …correct the sentence

Figure 2: no scale bar, no magnification mentioned

Figure 3: A, B, C, D, E missing but mentioned in Figure 3 legend.

Figure 4B:  letter for significant difference missing at MM

Author Response

Please see the attachment response to Reviewer 4 Comments

Round 2

Reviewer 3 Report

Point 1: The mycorrhizal fungi used for inoculation were not characterized, the only known information is a natural source, where the organisms were collected. It seems to be quiet surprising that the inoculation generally led to increase of height and leaf surface, but no significant increase of shoot weight nor seedlings total weight was observed. Is there some explanation for that? This should be addressed in discussion. Although some significant differences were presented between the inoculated and uninoculated plants in their content of tannins, flavonoids or total phenolic, the correlation between the mycorrhiza and phytochemicals seems to be very weak (lines 233-234). Moreover, the used methods of measurements of flavonoids and total phenolic are rather a general estimation, allowing just preliminary conclusions.

Response 1: We observed an increase in shoot weight and total weight, albeit not significant (p=0.64 and p=0.77, respectively). In contrast, seedlings height and leaf surface were significantly higher (p=0.0083 and p=0.0059, respectively) compared to the control, when the seedlings were inoculated with AMF (Figure 5). This could be due to the early stage association between AMF and seedlings, which may involve an unbalanced bidirectional exchange of carbon transfer and nutrients during the early stages of growth [45,46].

Advanced methods of phytochemical content measurement such as High Performance Liquid Chromatography (HPLC) could be used in downstream investigation for more and accurate information.

Response of reviewer: According to the growth; Well, but even regardless to the significance (OK, significance is important, but for simplification), just focus to the figure 6: If you compare the control with MC - they have very similar height and leave surface but MC has more shoot weight. On the other hand MK and MM are higher and have more leave surface (more leaves?) but their shoot weight in lower than MC although somewhat higher than in control. I seems like that the height and leave surface would not be very relevant for the total shoot weight (this is what is surprising to me)? What is affecting the shoot weight more than number of leaves and height? Thickness of stem (trunk), leaves? Or what? This is what I do not understand.

In general, as I wrote, the presented results are a good foundation, but further studies would be necessary to increase the relevance of the manuscript. Authors basically agreed with this opinion. Up to my opinion, somewhat more relevant results are needed for a manuscript suitable for publication in journal Plants. (It was not improved in that point.)

Reviewer 4 Report

The Authors reply to my comments sounds relevant.